# Statistics of pathogenic bacteria in the search of host cells

Stefan Otte[1,2,6], Emiliano Perez Ipiña [1,2,3,6], Rodolphe Pontier-Bres[2,4], Dorota Czerucka [2,4✉] & Fernando Peruani [1,2,5✉]

A crucial phase in the infection process, which remains poorly understood, is the localization of suitable host cells by bacteria. It is often assumed that chemotaxis plays a key role during this phase. Here, we report a quantitative study on how *Salmonella* Typhimurium search for T84 human colonic epithelial cells. Combining time-lapse microscopy and mathematical modeling, we show that bacteria can be described as chiral active particles with strong active speed fluctuations, which are of biological, as opposed to thermal, origin. We observe that there exists a giant range of inter-individual variability of the bacterial exploring capacity. Furthermore, we find *Salmonella* Typhimurium does not exhibit biased motion towards the cells and show that the search time statistics is consistent with a random search strategy. Our results indicate that in vitro localization of host cells, and also cell infection, are random processes, not involving chemotaxis, that strongly depend on bacterial motility parameters.

[1] Université Côte d'Azur, Laboratoire J.A. Dieudonné, UMR 7351 CNRS, Nice, France. [2] LIA ROPSE, Laboratoire International Associé Université Côte d'Azur - Centre Scientifique de Monaco, Monaco, Monaco. [3] Department of Physics & Astronomy, Johns Hopkins University, Baltimore, MD, USA. [4] Centre Scientifique de Monaco (CSM), Monaco, Monaco. [5] Laboratoire de Pysique Théorique et Modélisation, UMR 8089, CY Cergy Paris Université, Cergy-Pontoise, France. [6] These authors contributed equally: Stefan Otte, Emiliano Perez Ipiña. ✉email: dczerucka@centrescientifique.mc; fernando.peruani@cyu.fr

Gastrointestinal infections by pathogenic, flagellated bacteria, such as *Salmonella enterica* serovar typhimurium (ST) involve multiple steps. The infection starts with bacteria performing three-dimensional movements in the gut lumen until reaching specific areas inside the gastrointestinal tract[1]. The next phase requires bacteria to approach the intestinal epithelium, move on it, to finally reach and dock on host cells (HCs)[2]. The final step of the invasion in ST involves the internalization inside the HCs[2,3]. After anchoring to the cell membrane, bacteria make use of the Type III secretion system[4] to trigger a series of complex processes that culminate by the formation of an endocytic vesicle that allow bacteria to enter into the cell. Our understanding of the role played by bacterial motility in the infection process is very limited. Experiments with mice with motile and non-motile bacteria have provided evidence that active motility facilitates infection[5,6]. In line with these results, in vitro experiments with HCs attached to abiotic surfaces have shown that cell invasion is enhanced by active motility[7–9]. This observation is also consistent with results obtained on biotic (cellular) surfaces[2]. Furthermore, recent studies performed with gut explants have shown that active motility becomes essential to breach the mucus layer, as well as to locate mucus-free areas (e.g., the cecum in mice) in order to obtain direct access to the tissue[1]. How bacteria navigate towards these suitable areas of the gastrointestinal tract and identify host cells remains poorly understood. Specifically, there is no quantitative understanding of the bacterial search strategy. Though chemotaxis is assumed to play a key role, it has been found that it is not essential for infection on glass surfaces or tissue culture cells[2], but it is required to promote the infection in the gut[5,6].

Surfaces are of fundamental physiological relevance for bacteria: the level of nutrients is higher near them and host cells sit off them, when it is not that the surface itself is a cellular tissue prone to be infected. Bacterial swimming patterns near surfaces are fundamentally different from those observed in the bulk[10–20]. Near a surface, bacteria experience an effective, hydrodynamic-induced attraction towards the surface[10–14], tumbling events are strongly suppressed[21,22], and hydrodynamic-induced torques[18–20,23] force bacteria to move in circular trajectories[10–14,24,25]. Furthermore, depending on the properties of the surface and bacterial adhesins, optimal surface exploration requires bacteria to perform transient adhesion events at a given frequency[26]. For too strong or too weak adhesion properties, adhesion events are detrimental for surface exploration[26], as occurs for ST on the glass.

Here, we present a quantitative study that combines time-lapse microscopy and active matter modeling to characterize the motion of pathogenic bacteria in the search of HC. We use ST as pathogenic bacteria and T84 human colonic epithelial cells as HCs. We show that near-surface motion of ST corresponds to that of a chiral active particle, characterized by an angular speed $\Omega$, with an important distinctive feature: ST exhibits strong active speed fluctuations. We provide evidence that the observed speed fluctuations are of biological, as opposed to thermal, origin. Furthermore, we show that in the absence of transient surface adhesion, active speed fluctuations play a key role in the exploration capacity—i.e., diffusion coefficient—of these bacteria, which can account for up to 40% of its value when $\Omega > 1\,\mathrm{s}^{-1}$. In addition, we find that there exists a large inter-individual variability—over four orders of magnitude—of exploration capacity within the population of motile bacteria. Our analysis indicates that ST does not exhibit chemotaxis towards the HCs. Furthermore, we show that the statistics of encounter times between ST and HC—hitting-time statistics—are fully consistent with a random search strategy (RST), implying that encounters are fortuitous events. We use this knowledge to discuss the relation between motility parameters and bacterial virulence.

## Results

We study first the motility of wild-type ST, strain SL1344 (from now on ST-WT) near the bottom glass coverslip of an invitrogen Attoflour chamber without HC, filled with 4 mm height liquid film of Dulbecco's Modified Eagle Medium (DMEM) at 37 °C using phase-contrast microscopy at 40× magnification (for further details, see Methods). Experiments with human T84 epithelial colonic cells (HC) are performed under identical conditions. In order to determine whether chemotaxis is used by ST-WT, we perform control experiments with non-chemotactic ST mutants—strain M935 (from now on ST-M935)—in the presence of HC. In order to determine the level of bacterial infection, we report results from infection experiments in which the number of bacteria internalized inside HCs is estimated. These experiments are performed with ST-WT as well as a control experiment with non-flagellated mutants, strain M913 (from now on ST-M913). Estimates are provided 60 min after inoculation of bacteria (for further details, see Methods).

**Bacterial motion in the absence of host cells.** We start out by analyzing bacterial trajectories in the absence of HC. Note that the bacterial population, as observed in vivo and in vitro experiments, includes flagellated as well as non-flagellated bacteria owing to phenotypic noise[27]. Non-flagellated bacteria perform passive diffusion dominated by thermal and environmental fluctuations, thus they are inefficient to explore space and locate HCs. Here, we focus on flagellated-driven bacteria and analyze their capacity to explore the environment and locate HCs. For an illustration of trajectories of actively moving bacteria, see Fig. 1a–d and Supplementary Movies 1–4. We proceed as follows. For each bacterium, we compute the position of its center of mass, with a positional error estimate of 0.3 μm, every $\Delta t = 0.03$ s, with $\Delta t$ the time between two consecutive frames, and construct $\Delta \mathbf{x}_{i,n} = (\Delta x_{i,n}, \Delta y_{i,n}) = (x(t_i + n\Delta t) - x(t_i), y(t_i + n\Delta t) - y(t_i))$, where $t_i$ refers to the time associated to frame $i$, and $n$ is a non-negative integer. From $\Delta \mathbf{x}_{i,n}$, we obtain the velocity vector $\mathbf{V}_{i,n} = \Delta \mathbf{x}_{i,n}/(n\Delta t)$ and extract the speed $u_{i,n} = \sqrt{\Delta x_{i,n}^2 + \Delta y_{i,n}^2}/(n\Delta t)$; see Supplementary Fig. 1. Note that $\mathbf{V}_{i,n}$ and $u_{i,n}$ are average quantities, even for $n = 1$, of an underlying continuous-time process. For instance, one can express $\mathbf{V}_{i,n} = (n\Delta t)^{-1}[\int_{t_i}^{t_i+n\Delta t} \dot{\mathbf{x}}(s)ds + \mathbf{E}_{i,n}(t)]$, where $\dot{\mathbf{x}}$ the instantaneous velocity in the underlying continuous process and $\mathbf{E}_{i,n}(t)$ is the experimental error performed in the velocity estimation. It is worth stressing that $\dot{\mathbf{x}}$ and $\mathbf{E}_{i,n}$ are stochastic independent variable, $\mathbf{E}_{i,n}$ is independent of $n$ ($\mathbf{E}_{i,n} = \mathbf{E}_i$), whereas the variance of $\int_{t_i}^{t_i+n\Delta t} \dot{\mathbf{x}}(s)ds$ grows with $n$. As result of this, as explained in detail in Supplementary Note 2, it is possible to discriminate for $n > 1$ contributions to fluctuations coming from experimental errors from those genuinely related to the underlying physical/biological process, in order to obtain a reliable characterization of speed fluctuations. Although we use $n > 1$ to obtain accurate estimates, note that the observed speed fluctuations (over time) are significantly larger than the experimental error ($\pm 10$ μm/s) even for $n = 1$ [Fig. 1e]. In the following, to ease the notation we refer to the average velocity as $\mathbf{V}(t)$, to its magnitude as $u(t)$, and drop the index $n$ and use continuous-time $t$ instead of $i$. We find that $u(t)$ is normally distributed [Fig. 1f], whereas the speed autocorrelation $A(t)$ [Fig. 1g] takes the form:

$$A(t) = \frac{\langle (u(t') - \bar{v})(u(t' + t) - \bar{v}) \rangle_{t'}}{\langle (u(t') - \bar{v})^2 \rangle_{t'}} = e^{-k_v t}, \quad (1)$$

where $k_v$ is the time-relaxation constant, $\bar{v}$ denotes the average (along the trajectory) of the speed, and $\langle \cdots \rangle_{t'}$ indicates average

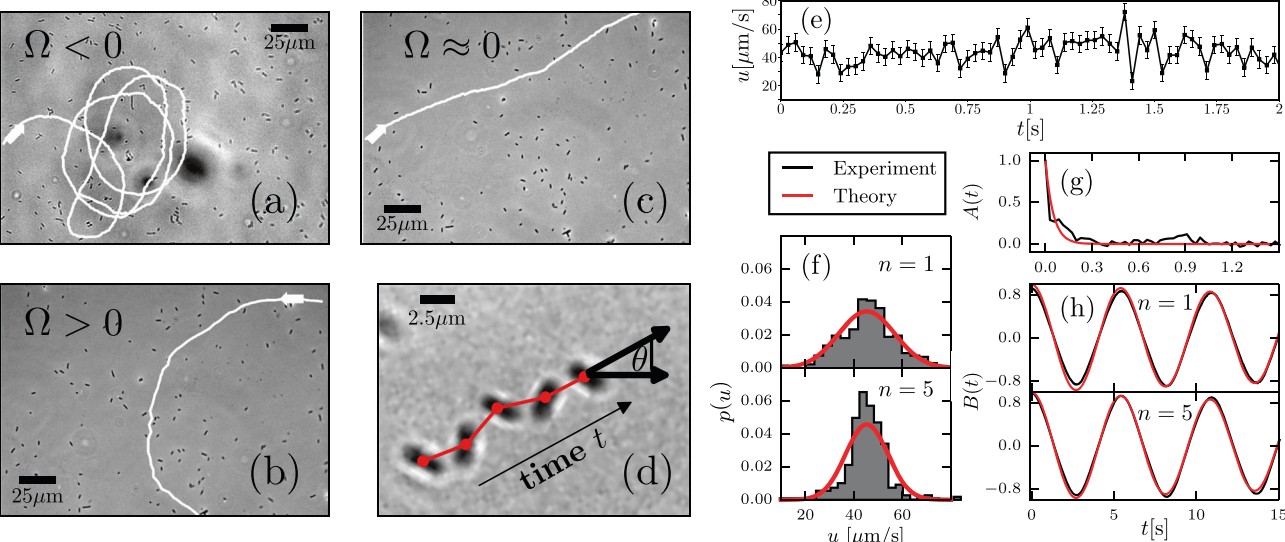

**Fig. 1 Experimental trajectories in the absence of host cells (HCs) and estimation of motility parameters.** In **a–c**, we chose in each of them one motile bacterium, with a distinct angular speed $\Omega$, from a population that includes motile as well as non-motile bacteria, and display its trajectory. The different panels illustrate the presence of different types of bacterial trajectories. **d** At short time-scales we observe that the longest axis of the bacterium oscillates, however, the center of mass of the bacterium moves along smooth paths. **e** Speed fluctuations: speed $u$— computed using two consecutive frame, i.e., $n = 1$ —vs time in a representative trajectory. Error bars correspond to the experimental SE in the estimation of $u$, see details on the Supplementary Note 2. **f** Distribution of the speed $u$ computed for $n = 1$ and 5 values. The red curve corresponds to Eq. (4). **g** Speed correlations $A$, using $n = 1$. The red curve corresponds to Eq. (1). **h** Moving direction correlations $B$, averaging over $1\Delta t$ and $5\Delta t$. The red curve corresponds to Eq. (2). Note that correlations remain unchanged for $n = 1$ and $n = 5$. For further details, see Supplementary Note 1 and 2, as well as Supplementary Fig. 1.

over $t'$. The autocorrelation of the moving direction—defined as $\mathbf{e}(t) = \frac{\mathbf{V}(t)}{u(t)}$—exhibits the following functional form [Fig. 1h]:

$$B(t) = \langle \mathbf{e}(t') \cdot \mathbf{e}(t' + t) \rangle_{t'} = e^{-D_\theta t} \cos(\Omega t) , \quad (2)$$

where $\Omega$ is a constant frequency (or angular speed) and $D_\theta$ a time-relaxation constant.

The speed and moving direction correlations given by Eqs. (1) and (2), respectively, are consistent with the following equations of motion for chiral active particles with active speed fluctuations:

$$\dot{\mathbf{x}}(t) = v(t)\hat{e}(\theta(t)) \quad (3a)$$

$$\dot{\theta}(t) = \Omega + \sqrt{2D_\theta}\xi_\theta \quad (3b)$$

$$\dot{v}(t) = -k_v(v - \bar{v}) + \sqrt{2D_v}\xi_v \quad (3c)$$

where dots refer to temporal derivatives, $\mathbf{x}$ is the position of the center of mass of the bacterium, $\mathbf{e}(\theta(t)) = (\cos\theta(t), \sin\theta(t))$ is a unit vector that indicates the bacterium moving direction, and $v(t)$ corresponds to the instantaneous active speed. Recall that $u(t)$ refers to the average speed. Both noises $\xi_m(t)$ are delta-correlated such that $\langle \xi_m(t')\xi_m(t'') \rangle = \delta(t' - t'')$ for $m = \theta, v$. Eq. (3c) defines an Ornstein-Uhlenbeck process, whose steady state speed distribution $p(u)$ is then given by:

$$p(v) = \frac{1}{\sqrt{2\pi\frac{D_v}{k_v}}} \exp\left[\frac{-(v - \bar{v})^2}{2\frac{D_v}{k_v}}\right] . \quad (4)$$

The estimation of $k_v$ is obtained from Eq. (1), whereas $\bar{v}$ and the amplitude of speed fluctuations $D_v$ are estimated by the procedure detailed in Supplementary Note 2. Finally, the rotation frequency $\Omega$ and the amplitude of moving direction fluctuations $D_\theta$ are obtained from Eq. (2). A detailed derivation of Eqs. (1) and (2) from Eqs. (3) is provided in Supplementary Note 1 and further details on the estimation of parameters can be found in Supplementary Note 2. Note that assumption of an OU process in the speed allows us to take into account active (as well as

thermal) speed fluctuations[28–30]). To the best of our knowledge, this is the first model of active chiral particles with active speed fluctuations; in the absence of active fluctuations, the model reduces to the one studied in[31–34].

The analysis of the model given by Eq. (3) reveals various remarkable features of chiral active particles with active speed fluctuations. The first moment, $\langle \mathbf{x}(t) \rangle$, in the presence of moving direction fluctuations is an inward spiral that asymptotically converges to a point, as previously reported in ref. [33]. In the absence of such fluctuations, it becomes, due to active speed fluctuations, in sharp contrast to the previous scenario, an outward spiral that asymptotically converges to a limit cycle, see insets in Fig. 2a and b; the expression and derivation of $\langle \mathbf{x}(t) \rangle$ are given in Supplementary Note 1, see Supplementary Eq. (14). From the study of $\langle \mathbf{x}^2(t) \rangle$, we learn that when both fluctuations are present, i.e., $D_v > 0$ and $D_\theta > 0$, the mean-square displacement displays decaying oscillations mounted on a linearly increasing function of time, Fig. 2b; the expression and derivation of $\langle \mathbf{x}^2(t) \rangle$ is given in Supplementary Note 1, see Supplementary Eq. (17). We find that even in the absence of moving direction fluctuations ($D_\theta = 0$) the mean-square displacement grows with time, Fig. 2a. Finally, the active diffusion coefficient $D$ of these particles is given by:

$$D = \frac{\bar{v}^2 D_\theta}{2(D_\theta^2 + \Omega^2)} + \frac{D_v}{2k_v}\frac{(k_v + D_\theta)}{((k_v + D_\theta)^2 + \Omega^2)} , \quad (5)$$

where the second term, $D_B = \frac{D_v}{2k_v}\frac{(k_v + D_\theta)}{((k_v + D_\theta)^2 + \Omega^2)}$, is exclusively due to the presence of speed fluctuations and reduces to $D_B = D_v/(2k_v^2)$ for $k_v \gg D_\theta, \Omega$. For details on the derivation of Eq. (5), see Supplementary Note 1. We find that $D$ grows with speed fluctuations ($D_v$), Fig. 2c, decreases with $\Omega$, Fig. 2d, and exhibits a maximum with moving direction fluctuations ($D_\theta$), Fig. 2e. It is worth stressing that while speed fluctuations are typically negligible for non-chiral active particles, here we show that for chiral active particles, active speed fluctuations can have a

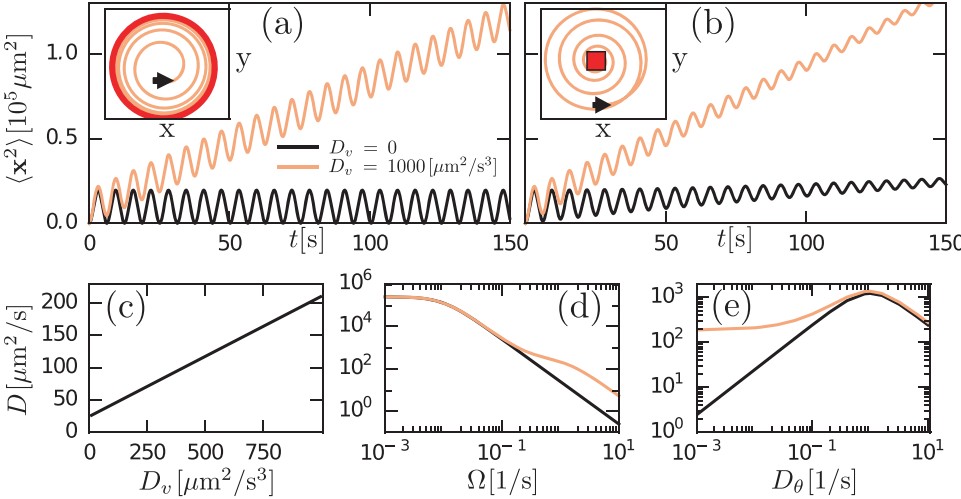

**Fig. 2 Theoretical prediction for a two-dimensional chiral active particle with active speed fluctuations.** In **a** and **b** it is shown that both $\langle \mathbf{x}(t) \rangle$ (inset) and $\langle \mathbf{x}^2(t) \rangle$ (main panel) evolves in time. The presence ($D_v > 0$) or absence ($D_v = 0$) of speed fluctuations play a different role for $D_\theta = 0$ (**a**) and $D_\theta > 0$ (**b**). In particular, $\langle \mathbf{x}(t) \rangle$ is an outward spiral that converges to a limit cycle [red circle in inset of **a**], whereas in the presence of such fluctuations we observe an inward spiral converging to a point [red square in inset of **b**]. **c–e** display the effective diffusion coefficient $D$ as function of $D_v$, $\Omega$, and $D_\theta$, respectively. **d** and **e** show $D(\Omega)$ and $D(D_\theta)$ in the presence (orange) and absence (black) of speed fluctuations.

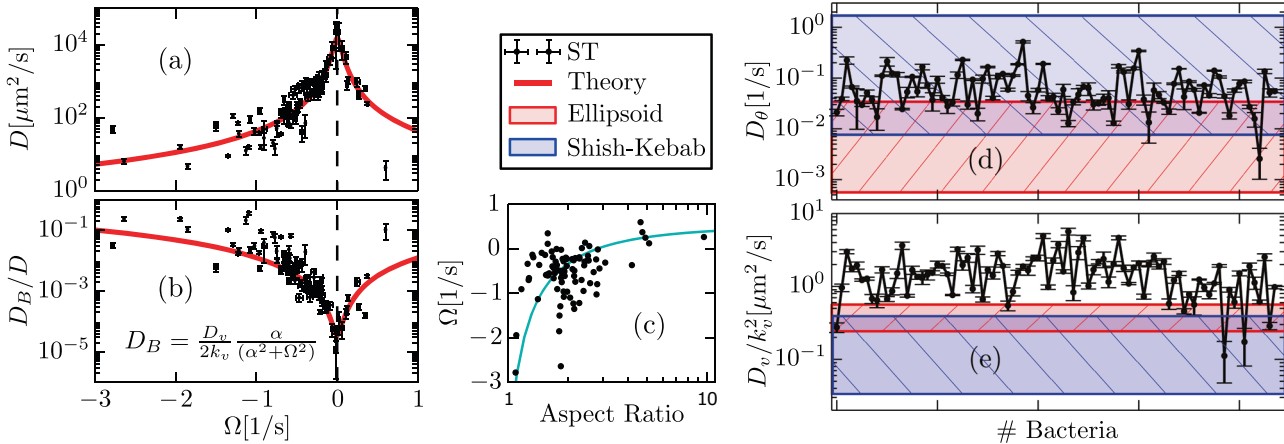

**Fig. 3 Variability of individual diffusion coefficients and origin of fluctuations. a** Individual diffusion coefficient $D$ of ST-WT as a function of $\Omega$. The symbols correspond to experimental estimations and the red curve corresponds to $D_{AV}(\Omega)$, whose expression is given in the text. The relative importance of speed fluctuations, given by $D_B/D$ to the estimation of $D$ is shown in **b**; $\alpha = D_\theta + k_v$. There seems to exist a correlation between $\Omega$ and the aspect ratio $\omega$ of bacteria as indicated in **c**; the solid curve is a guide to the eye. **d** and **e** show the measured values for $D_\theta$ and $D_v/(k_v^2)$, respectively, and compares them with the expected range of values for thermal fluctuations using the so-called ellipsoid (red color) and Shish-Kebab (blue color) model[38,39], see text and Supplementary Note 4. All error bars correspond to SE and were computed by Gaussian propagation of errors.

measurable and significant impact in the transport properties of these particles, including in the diffusion coefficient, provided $\Omega$ is large enough (in bacteria, see below, for $\Omega > 1\,\mathrm{s}^{-1}$).

Figure 3a shows that the diffusion of individual, flagellar-driven ST bacteria swimming near a surface ranges from 1 to $10^4\,\mu\mathrm{m}^2/\mathrm{s}$. Note that though we analyze here flagellar-driven bacteria only, diffusion coefficient values of the order of $1\,\mu\mathrm{m}^2/\mathrm{s}$, precisely 0.2 $\mu\mathrm{m}^2/\mathrm{s}$, are similar to those expected for non-flagellated bacteria driven by thermal Brownian motion[1]. On the other hand, diffusion coefficients of the order of $100\,\mu\mathrm{m}^2/\mathrm{s}$, observed for $|\Omega| = 1/\mathrm{s}$, coincide with those for flagellated bacteria in the near-surface colonic mucus layer[1]. Such a large inter-individual variability of diffusion coefficients—for flagellated bacteria in the same medium—is a direct consequence of the nonlinear functional form of $D$, Eq. (5), and its functional dependency with $\Omega$, which varies in the range $-3\,\mathrm{s}^{-1} < \Omega < 1\,\mathrm{s}^{-1}$. To illustrate this fact, we compute the diffusion coefficient $D_{AV}(\Omega)$ of a

representative bacterium, whose motility parameters, but the rotation frequency $\Omega$, correspond to the average values obtained using all-analyzed bacterial trajectories, that we express, using Eq. (5) in the limit $k_v \gg D_\theta, \Omega$, as $D_{AV}(\Omega) = \frac{\langle \bar{v} \rangle^2 \langle D_\theta \rangle}{2(\langle D_\theta \rangle^2 + \Omega^2)} + \frac{1}{2} \langle \frac{D_v}{k_v^2} \rangle$, with mean values: $\langle \bar{v} \rangle = 39.45\,\mu\mathrm{m}\;\mathrm{s}^{-1}$, $\langle D_v/k_v^2 \rangle = 2.38\;\mathrm{m}^2\mathrm{s}^{-1}$, and $\langle D_\theta \rangle = 0.076\;\mathrm{s}^{-1}$. The diffusion coefficient $D_{AV}(\Omega)$ is shown by a solid red curve in Fig. 3a and b. For further details on the statistics of the trajectories, see Supplementary Note 3 as well as Supplementary Figs. 2 and 3. To assess the contribution of active speed fluctuations to the diffusion coefficient, we compare in Fig. 3b the value of $D_B$ with respect to $D$. We find that for $|\Omega| > 1$ the estimated value of the $D$ can only be explained by considering speed fluctuations, which can account for up to 40% of the actual value. On the other hand, for vanishing values of $|\Omega|$ we observe the largest $D$, with $|\Omega| \to 0$ not diverging but corresponding to the $D$ of a non-chiral active particle subject to fluctuation in the

moving direction. Note that a correlation between the aspect ratio $\omega$ of a bacterium and its measured $\Omega$ value seems to exist, Fig. 3c. This observation seems consistent with arguments put forward in ref. [35], however other explanation[33,36] may provide alternative explanations.

As indicated above, details on experimental fluctuation estimations are provided in Supplementary Note 2. Specifically, measurements of speed fluctuations are performed by studying displacement fluctuations in time intervals $n\Delta t$ with $5 < n \leq 7$, making use of Supplementary Eq. (18) and the method explained in Supplementary Note 2; see Supplementary Fig. 1c. The developed procedure allows accurate and reliable measurements of $D_v/k_v^2$, and thus of speed fluctuations. In the following, we focus on the possible origin of the observed fluctuations. In the analysis, we ignore the fact that thermal fluctuations in $\hat{z}$ lead to changes in drag coefficients, thus contributing effectively to $D_\theta$ and $D_v$. However, fluctuations induced by thermal motion in $\hat{z}$, in near-surface swimming, lead to correction smaller than the one found in our experiments[37]. If fluctuations are of thermal origin, then we expect to observe that $D_\theta$ is proportional to $K_BT/\zeta_R$ and measurements of $D_v/k_v^2$ to $K_BT/\zeta_0$, where $K_B$ is the Boltzmann constant, $T$ the temperature (in Kelvin), and $\zeta_R$ and $\zeta_0$ drag constants associated to rotations and displacements of the bacterium, respectively. Note that there exist several drag friction coefficient models for spherocylinders that depend on the viscosity of the liquid, length, and radius of the object. We use the so-called Shish-kebab and ellipsoid models[38,39] for bulk drags, which provide an upper bound for thermal fluctuation estimates. Figure 3d shows the estimated values of $D_\theta$ and compares them with the expected values of these two models, using the maximum and minimum measured bacterial aspect ratio. The figure indicates that measured

values of $D_\theta$ are consistent with thermal fluctuations at 37 °C. Moreover, the measured value $\langle D_\theta \rangle \approx 0.076\ s^{-1}$ is remarkably close to the early estimates of thermal rotational diffusion for bacteria by Berg[15]. Performing the same exercise for $D_v/k_v^2$, which has dimensions of spatial diffusion coefficient, we find, Fig. 3d, that the measured values are systematically higher than the expected values of both Shish-kebab and ellipsoid model. For further details, see Supplementary Note 4 and Supplementary Movie 4. This finding indicates that speed fluctuations are not consistent with thermal fluctuations. The observed higher values, we speculate, result from the combined effect of thermal fluctuations and fluctuations coming from the swimming propulsion machinery, likely to result from a non-trivial interplay of fluctuations of the individual flagella forming the bundle[40,41].

**Bacterial motility in the presence of host cells.** In the following, we study the behavior of ST in the presence of HC (see Supplementary Movie 5). The comparison of experiments with (i) ST-WT in the absence of HC, (ii) ST-WT in the presence of HC, and (iii) the non-chemotactic mutant ST-M935 in the presence of HC allows us to provide solid evidence that ST-WT does not display biased motion towards HC, see Fig. 4a–c. In order to do that, we define for each bacterium the quantity $l_{min}$ that measures the distance (from the bacterium) to the closest HC. The temporal evolution of $l_{min}$ for an experimental trajectory is shown in Fig. 4d. The sign of the temporal derivative of $l_{min}$, $m(t) = \frac{dl_{min}}{dt}$, indicates whether the bacterium is approaching or moving away from the closest HC at time $t$, see Fig. 4e. We use the temporal average of $m(t)$ for a trajectory—denoted by $\bar{m}$—as "bias" order

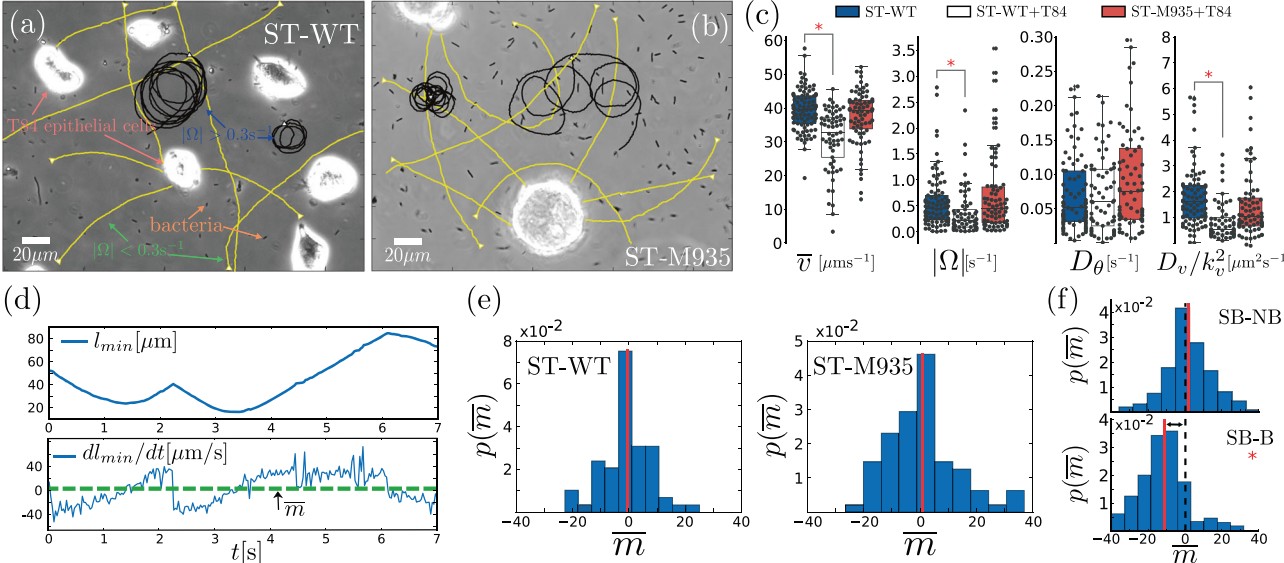

**Fig. 4 Experimental trajectories in the presence of HC. a** and **b** show snapshots of experiments with ST-WT (**a**) and non-chemotactic ST-M935 (**b**) (black dots) in the presence T84 epithelial cells as HC (bright areas). Some representative bacterial trajectories—of a population that includes motile and non-motile bacteria—are indicated by solid lines: yellow for $|\Omega| < 0.3\ s^{-1}$ and black for $|\Omega| > 0.3\ s^{-1}$. The classification of trajectories in two categories, those with $|\Omega|$ smaller and larger than $0.3\ s^{-1}$, is simply to highlight the importance of $|\Omega|$. **c** Box plots of the (average) motility parameters in experiments: (i) with ST-WT in absence of T84 cells (ST-WT, $n = 89$), (ii) with ST-WT in the presence of T84 cells (ST-WT+T84, $n = 60$), and (iii) with the non-chemotactic mutant ST-M935 in the presence of T84 cells (ST-M935+T84, $n = 75$). **d** illustrates the temporal evolution of the distance $l_{min}$ to the closest T84 cell and its derivative, $m(t) = dl_{min}/dt$ for a representative bacterial trajectory; the average of $m(t)$, which is our "bias" order parameter ($\bar{m}$) is shown as a dashed horizontal line. **e** The distribution of $\bar{m}$ in experiments, with ST-WT (left) and ST-M935 (right), in the presence of T84 cells. The vertical red lines indicate the population average. **f** Same as **e**, the method is illustrated using simulated data with (SB-B) and without (SB-NB) a chemotactic bias. The black dashed line indicates $\bar{m} = 0$ as a reference. Box plots: the lower and upper limit of the boxes correspond to the 25th and 75th percentiles. The line inside indicates the median. The upper and lower whiskers extend values within 1.5 IQRs. The red asterisk indicates $p$ value < 0.01. See Supplementary Table 1 for details about the statistical tests.

parameter. For an ensemble of trajectories, in absence of biased motion, the (population) average of $\bar{m}$ is expected to be centered about 0: half of the trajectories are approaching cells, whereas the other half is moving away. The presence of biased motion toward HCs introduces a shift of the average of $\bar{m}$ towards negative values. This observation can be easily verified in simulations with and without chemotactic bias towards HCs [Fig. 4e]. The method applied to experiments with ST-WT and with non-chemotactic mutant ST-M935 in the presence of T84 cells shows that for both, wild-type and mutant, the distribution of $\bar{m}$ is centered ~0. A binomial test, using the sign of $\bar{m}$, yields in experiments with ST-WT a $p$ value = 0.60, whereas in experiment with ST-M935, $p$ value = 0.99. This indicates that the null hypothesis—absence of chemotactic bias, for which the probability of observing for a trajectory $\text{sgn}(\bar{m}) > 0$ is 0.5—is consistent with the data. Thus, this proves that a chemotactic bias is neither present for non-chemotactic mutants nor for wild-type ST. Other tests, included in Supplementary Note 5, lead to the same conclusion. Note that motility parameters between ST-WT without HC and ST-M935 do not display a significant difference, whereas those for ST-WT in the presence of HC are slightly modified, see Fig. 4c and Supplementary Table 1. For further details on biased motion tests, see Supplementary Note 5 as well as Supplementary Figs. 4–6.

We study the statistics of search times—i.e. the time that takes for a bacterium to reach an HC—which we characterize by the (cumulative) probability $S(\tau)$ that this time is larger or equal to $\tau$. Figure 5a shows $S(\tau)$ in experiments with ST-WT and with the non-chemotactic ST-M935 in the presence of HC, as well as

in a control statistical test. The control test consists of using trajectories of ST-WT in absence of HC to compute the time these trajectories hit randomly distributed areas that mimic the presence of HC (for details, see Supplementary Note 6); the distribution of positions and sizes of HCs using in the test correspond to those in experiments with HC (Supplementary Note 7 and Supplementary Fig. 7). The analysis reveals that $S(\tau)$ is almost identical for the wild-type (ST-WT), the non-chemotactic mutant (ST-M935), and the control test. These results support the finding that the encounter of ST and HC does not involve biased motion towards HCs and results from a random process. All this indicates that the distribution of $\tau$ can be estimated as the first-passage time of the random chiral walker defined by Eq. (3), and thus, one expects the statistics of $\tau$ to strongly depend on mobility parameters, specially $\Omega$; Supplementary Fig. 7 shows $\langle\tau\rangle$ vs. $\Omega$. Figure 5b and c show that in experiments, $\langle\tau\rangle$ is larger when $|\Omega| > \Omega_0$ than when $|\Omega| < \Omega_0$, which confirms the sensitivity of $\langle\tau\rangle$ to the motility parameter $\Omega$. In the figure, we have used $\Omega_0 = 0.3\,s^{-1}$ to illustrate this effect, but any other value of $\Omega_0$ can be used. The conclusion is always that bacteria with $|\Omega| > \Omega_0$ require in average more time to find a HC than those with $|\Omega| < \Omega_0$. Under the condition $\bar{v}\Omega^{-1} \ll 1/\sqrt{\rho_{HC}}$, with $\rho_{HC}$ the host cell density, $\langle\tau\rangle$ can be roughly estimated as $\langle\tau\rangle \sim [4\rho_{HC}D(\Omega)]^{-1}$[42], otherwise the encounter of ST and HC is given by a "ballistic" regime that does not depend on $D$. This is illustrated in Fig. 5d that displays $\langle\tau\rangle$ as function of $D$ in experiments (symbols) and in simulations of the active chiral particle model for various $\Omega$ values, using for other motility parameters the (population) average values. Such a

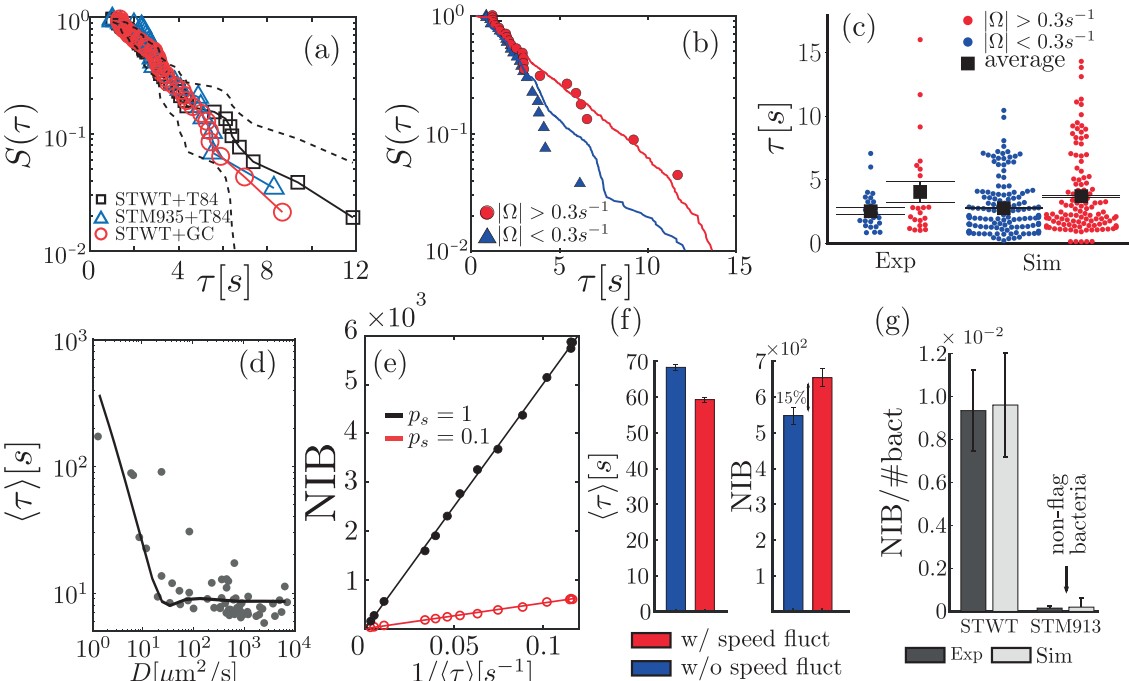

**Fig. 5 Search time statistics and infection process. a** Probability $S(\tau)$ that the search time is larger than $\tau$ in experiments with ST-WT ($n = 50$) and ST-M935 ($n = 28$), respectively, in the presence of T84 cells and in a control test (STWT+GC, $n = 63$), see text. The dashed curve represents the 95% confidence interval of $S(\tau)$ for ST-WT. **b** $S(\tau)$ for ST-WT with $|\Omega| > 0.3\,s^{-1}$ (red) and $|\Omega| < 0.3\,s^{-1}$ (blue) in experiments (symbols, $n = 27$ and $n = 23$) and simulations (curves, $n = 1445$ and $n = 1169$); in **c** the corresponding search time distribution and their average (black square) $\langle\tau\rangle$ is shown. **d** $\langle\tau\rangle$ as function of $D$: the solid curve was computed using a representative bacterium whose motility parameters correspond to average values, except for $\Omega$ that is used to vary $D$; dots correspond to estimates using individual motility parameters. **e** Number of invading bacteria (NIB) vs $1/\langle\tau\rangle$ for two infecting probabilities: symbols correspond to simulations and straight lines to linear fits. **f** $\langle\tau\rangle$ and NIB with (w/) and without (w/o) speed fluctuations for $\Omega = 2.5\,s^{-1}$ in simulations. **g** Percentage of NIB with respect to the total number of bacteria (#bact) calculated in experiments (dark gray, $n = 3$)and in simulations (light gray) with ST-WT and with non-flagellated mutant ST-M913; see Methods and Supplementary Note 7. Error bars in **c**, **f**, and **g** correspond to the standard error of the mean (SEM) for $\langle\tau\rangle$ and standard error (SE) for NIB.

relation between $\langle\tau\rangle$ and $D$ does not apply at high bacterial concentration—bacterial motion in these conditions can turn out to be superdiffusive[43]—or in the presence of an external flow[44].

We make use of our knowledge on the search time to simulate infections as follows. We first randomly distribute HCs over the space (at densities and size distributions as those in the experiments) and then integrate Eq. (3) to obtain bacterial trajectories. Whenever a bacterium encounters an HC, with probability $p_s$ the bacterium invades the HC, i.e., gets inside the HC, otherwise continues exploring the space. We compute, for a fixed time, the number of invading bacteria (NIB), i.e. the bacteria that managed to invade HC. Figure 5e shows NIB as function of $1/\langle\tau\rangle$, which provides a clear indication of the relevance of $\langle\tau\rangle$ in the infection process, with NIB inversely proportional to $\langle\tau\rangle$. All this suggests that the large range of individual $D$ values within the same population (cf. Fig. 3) may also imply inter-individual virulence variability. Finally, in order to assess the role of speed fluctuations in the computation of $\langle\tau\rangle$ and in the infection process, i.e., in the estimation of NIB, we compute both quantities in simulations that include speed fluctuations, as well as in simulations that neglect them, for values of $\Omega$ in the range $0$–$5\,\mathrm{s}^{-1}$ and using for other motility parameters, the average values obtained in ST-WT experiments. We find that for $\Omega \geq 1$, speed fluctuations lead to statistically significant increments of NIB (and decrease of $\langle\tau\rangle$). This is illustrated in Fig. 5f for $\Omega = 2.5\,\mathrm{s}^{-1}$, where the contribution due to speed fluctuations represents a 15% increase in NIB. For other values of $\Omega$, see Supplementary Fig. 10, and for further details on the implementation of the infection model and statistical tests, see Supplementary Note 8. Importantly, NIB can be directly estimated in experiments; for details see Methods. Finally, Fig. 5g compares results obtained in experiments[7] and in simulations. These results confirm that ST is actually invading HC and that motility plays an essential role in the infection process, as is evident from the low virulence exhibited by the non-flagellated mutants ST-M913[7]. The agreement between experiments and the infection model provides additional support to the relevance of motility in the infection process.

## Discussion

Through the performed quantitative analysis of ST motility patterns in absence of HC, we showed that ST can be described as chiral active particles with active speed fluctuations. The developed mathematical model of bacterial behavior allowed us to extract motility parameters and to compute the active diffusion coefficient ($D$) of individual flagellar-driven bacteria. It is worth stressing that given that bacteria, near the surface, behave as chiral swimmers, speed fluctuations contribute to the active diffusion coefficient for large enough values of the rotation frequency $|\Omega|$. Let us recall that for non-chiral particles, the contribution of active speed fluctuations to $D$ is too weak to be measurable. In the experiments, we found that active speed fluctuations can contribute up to 40% of the diffusion coefficient value, when $|\Omega| > 1\,\mathrm{s}^{-1}$. To the best of our knowledge, this is the first experimental example, where the contribution of active fluctuations to transport coefficients is measurable and significant. In addition, we showed that the measured speed fluctuations could not be of thermal origin, which led us to speculate that this phenomenon is related to fluctuations of the flagellar machinery[40,41]; further experiments, combining molecular biology and new microscopical techniques, may provide direct evidence of the correlation between speed fluctuations and the flagellar motor[45]. Furthermore, we observed that the obtained inter-individual variability of motility parameters lead to a diffusion coefficient that ranges over four orders of magnitude. We

presume that such variability among flagellar-driven motile bacteria is related to phenotypic noise, which has been shown in ST to play an important role in flagella synthesis and pathogenesis[27]. Note that bacterial population includes flagellated as well as non-flagellated bacteria[27], whereas here we provide evidence of the existence of large inter-individual variability among the sub-population of flagellar-driven motile bacteria.

In the presence of HCs, motility parameters are only slightly modified and the performed chemotaxis statistical tests indicate that ST does not exhibit biased motion towards cells, in line with the previous studies[2] that showed that in vitro chemotaxis is not essential for interaction with HCs. Nevertheless, note that in vivo chemotaxis is required to promote the infection in the gut[5,6]. The absence of chemotaxis has been also observed under some conditions, in which ST displays high motility and expresses virulence factors. This lets us speculate that HCs may induce the activation of type III secretion system without affecting ST motility[46,47].

The absence of biased motion towards HCs suggests that the time required for a bacterium to find an HC can be computed as a first-passage problem using the proposed chiral active model. The experimentally obtained search time statistics confirmed this idea, showing that bacterial behavior in the presence of HCs is analogous to the motion of active particles through a complex environment[48–51]. In short, our results indicate the strategy applied by bacteria to locate HCs corresponds to an RST. RSTs are commonly found in biology[52]. We speculate that a similar RST may be used by ST to find breaches in the mucus layer and reach the epithelium[1].

Finally, since the search time statistics can be understood as a first-passage time problem of active chiral particles, with a mean search time $\langle\tau\rangle$ function of the active diffusion coefficient $D$, we proposed a simple mathematical infection model that mechanistically relates motility parameters with infection capacity. Specifically, we argued that the number of invading bacteria (NIB) is determined by $\langle\tau\rangle$, and thus by $D$. In the light of this mathematical model, the large inter-individual variability found in $D$ translate into a large inter-individual variability of infection capacities among the subpopulation of flagellar-driven bacteria.

A comprehensive quantitative understanding of the bacterial infection process, and in particular of the role played by bacterial active motility, requires extensions of the here-developed motility model to characterize and describe bacterial motion within the mucus layer[1] and by passive transport in the lumen, as well as incorporating the complex interactions that take place when bacteria and host cells are in physical contact[2,7–9]. These issues should be the focus of future studies.

## Methods

**Bacteria**. *S. enterica* serovar Typhimurium (ST) strain SL1344 was kindly provided by Stéphane Méresse, Faculté des Sciences de Luminy, Centre d'Immunologie de Marseille-Luminy (CIML), INSERM-CNRS, Marseille, France. The non-chemotactic mutant strain M935 (*cheY*::Tn*10*) and the non-flagellated mutant strain M913 (*fliGHI*::Tn*10*) were kindly provided by Wolf-Dietrich Hardt from the Institute of Microbiology, D-BIOL, ETH Zurich, Switzerland[5]. Bacteria were stored in Luria-Bertani (LB) medium plus 15% glycerol at $-80\,^{\circ}\mathrm{C}$.

**Cell lines**. The human T84 colonic cell line was obtained from the European Collection of Animal Cell Cultures (Salisbury, England). The T84 culture medium contained a 1:1 mixture of Dulbecco-Vogt modified Eagle medium and Ham' s-F12 medium (DMEM/F12) supplemented with 50 µg/ml penicillin, 50 µg/ml streptomycin (Sigma, France), and 4% fetal bovine serum (Hyclone, France).

**Preparation for video-microscopy**. ST strain SL1344, which is a tumbling bacterial strain and the non-chemotactic mutant M935, were grown overnight into LB broth medium without shaking (a condition that preserved flagellum). Bacteria were pelleted by gentle centrifugation (1100 × g for 10 minutes) and resuspended in DMEM medium. For experiments without T84 cells, 2 ml of liquid was disposed

into invitrogen Attoflour cell chambers, leading to a density of ~$10^6$ bacteria/dishes. The circular cell chambers have a diameter of 25 mm, which leads to a height of ~4 mm of the liquid above the bottom glass surface of the cell chamber. For experiments with T84 cells, the cells were seeded at a density of $10^6$ cells/dish in a 35 mm glass-bottom dish (Mat Tek Corporation, USA). Twenty-four hours later culture medium was changed to medium without serum nor antibiotics for 12 hours. In order to perform infection, ST were added to cell chamber (~$5 \times 10^7$ bacteria/dishes). For time-lapse video-microscopy, the chambers were placed in a humidity (95%), $CO_2$ (5%), and temperature (37°C)—controlled environment. The focus was set on the coverslip at the bottom of the cell chamber, in order to record bacterial motion close to the surface glass/liquid. At least 30 minutes were given to the system to equilibrate prior to recording bacterial motion.

**Records of video-microscopy and bacterial tracking**. Motile bacteria were recorded by phase-contrast microscopy using a Leica DMI6000 B inverted microscope equipped with a high-sensitive Ropper CoolSnap HQ2 CCD camera (Photometrics) at ×40 magnification (numerical aperture: 0.75, Leica HCX PL Fluotar PH2). Images were acquired with the LAS-AF v4.0 software (Leica, Germany) at a rate of 35 fps. Images were 224.14 × 167.38 μm² (696 px × 520 px), 1 px ≈ 0.32 μm. Videos of bacterial motion were analyzed using the ImageJ platform v1.53c[53]. Trajectories of individual bacteria were tracked using MtrackJ v1.5.1 and the semi-automatic tracking tool from TrackMate v3.4.2 software[54,55]. Analysis of trajectories and figures were made using the algorithms detailed in Supplementary Note 2 implemented with Python 2.7.18 and Matlab R2016a.

**Infection procedure for invasion assays**. T84 cells were seeded into six-well tissue culture plates at $10^6$ cells per well. Twenty-four hours later, the culture medium was changed to medium without serum and antibiotics and maintained in this medium overnight. Infection was performed as described above for video microscopic procedure. Bacterial infection to T84 cells was quantified using the following method. After 1 h of infection, bacteria (outside cells) were eliminated by extensive washes with sterile phosphate-buffered saline (PBS). Cells were then incubated for an additional hour with DMEM/F12 containing 100 μg of gentamicin per ml. Since gentamicin was not concentrated in epithelial cells, intracellular bacteria survived the incubation, while adherent and extracellular bacteria were killed. The monolayers were then washed with sterile PBS, and epithelial cells with intracellular bacteria were detached by trypsin and lysed in water containing 0.1% bovine serum albumin. Different dilutions of the suspension were plated on LB-agar medium for colony-forming unit (CFU) number determination. CFU provides rough estimates of the number of bacteria contained inside T84 cells.

**Statistics and reproducibility**. Motility experiments were done $n = 3$ times. Images were captured from different cell chambers at different times. The field inside each cell chamber was chosen randomly. Tracked bacteria, larger than 0.2 sec were chosen randomly from different cell chambers and different experiments. For ST-WT in the absence of T84 cells, 89 independent trajectories were analyzed, for ST-WT and T84 cells, 60 trajectories, and for ST-M935 in the presence of T84 cells, 75 trajectories. Bacterial invasion experiments were replicated $n = 3$.

## Data availability
The raw data that support the findings of this study are available at http://data.centrescientifique.mc/Otte_data.html. Source data are provided with this paper.

## Code availability
The computer codes used for simulations and numerical calculations are available from the corresponding authors upon reasonable request.

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

## Acknowledgements

We thank L. Gómez Nava, R. Großmann, A. Koppler, M. Polin, and G. Volpe for insightful comments on the text. Experiments were performed at C3M Imaging Core Facility (Microscopy and Imaging platform Côte d'Azur, MICA) and simulations at CRIMSON clusters belonging to Observatoire Côte d'Azur. We acknowledge support from Agence Nationale de la Recherche via project *BactPhys*, Grant ANR-15-CE30-0002-01 and from Biocodex S.A., Gentilly, France.

## Author contributions

D.C. and F.P. designed the study. D.C. and R.P.-B. performed experiments. E.P.I., S.O., and F.P. performed the image and statistical analysis of the data and derived the mathematical models used to interpret the data. F.P. wrote the manuscript with the help of all authors.

## Competing interests

The authors declare no competing interests.
