## [Peer Review File · Nature Communications]

Reviewers' comments:

Reviewer #1 (Remarks to the Author):

Otte and colleagues have combined real time imaging, rigorous analysis of the bacterial tracks and modeling to study *Salmonella* swimming movement on solid surfaces and how this affects the infection process. In particular the modeling approach yields exciting new information about bacterial near surface swimming behavior. The spiral-like movement pattern and its dependence on speed fluctuations, directional fluctuations and the rotation speed provide important new angles for the field. *Salmonella* mutants are used to assess the role of chemotaxis and motility. These data are nicely in line with previous work and verify that in tissue culture infections, chemotaxis is not contributing to the host cell finding/infection. The present paper significantly extends the original observations by showing mathematically and by simulations that this pertains to both, directionality and search time. Overall, this is a very interesting and timely piece that will be of significant interest to the community. My comments may help the authors to improve the impact, i.e. in the infection biology community.

1. Fig. 1. It is surprising, that the authors do not show any non-flagellated (i.e. non-motile) cells, which are normally present *in vitro* and *in vivo* (PMID: 19096504 and PMID: 1824121). Indeed, in later Figures, such bacteria appear to exist also in their experiments. If omitted on purpose in Fig. 1, this should be explained to the reader.

2. Page 5. I wonder, if the fluctuations in speed may root in the fluctuations in motor assembly (PMID: 29971711 and PMID: 19234112). Also, previous data had shown fluctuations in flagellar motor speeds (PMID: 2407857 and PMID: 19234112). The authors may want to discuss this with relation to their data.

3. The abstract, the last paragraph of the introduction and the discussion section indicate that chemotaxis is not essential for the pathogen's interactions with glass surfaces or tissue culture cells. It should be pointed out to the readers, that this has been discovered, before (reference 1). Moreover, this also represents a key limitation of the *in vitro* plus modelling approach. In the gut, chemotaxis is indeed required to promote the infection (PMID: 18241212 and PMID: 15213159). For the infection biology field, it would be helpful to discuss the relevance of the observations for the *in vivo* infection.

4. A recent study has shown that *in vivo*, the pathogen in fact glides on the surface of cecum epithelial cells and on the surface of the gel-like mucus layer that covers large parts of the gut epithelium. On the mucus, the proposed random search strategy will actually help the pathogen to "find" breaches in the mucus layer, to reach the underlying epithelium (PMID: 31141690). I wonder, if this may be the main *in vivo* function of the mechanism described in the present work.

5. The original publication describing the *Salmonella* strains used should be cited (i.e. DOI: 10.1128/IAI.72.7.4138-4150.2004).

6. Fig. 3. In the gut, there is not only diffusing bacteria, but also some, that are indeed stuck to the mucus or the surface of epithelial cells (PMID: 31141690). How do the *in vitro* parameters fit to the *Salmonella* swimming parameters observed on intact gut tissue? Are any of the observed bacteria plotted in Fig. 3 stuck in a similar fashion? The authors may want to discuss how such an adhesion-mediated effect may affect the interpretation of the movement patterns with respect to flagellar function, alone.

7. The beginning of the discussion presents non-directional motility as an infection search strategy as an original discovery from this work. This should be put into context with previous work that has already demonstrated this experimentally (PMID: 22911370).

Reviewer #2 (Remarks to the Author):

This paper presents an experimental and theoretical study of the swimming of *Salmonella typhimurium* near glass surfaces and near surfaces of host cells. The authors first provide a quantitative characterization of the swimming and then ask whether host cells are reached (and subsequently invaded) randomly or by some type of directed/biased motion. Their analysis shows that motion to reach host cells is random, but dependent on motility.

Overall, this is a nice study, which tries to balance interesting physics with interesting biology. I

don't think it is a breakthrough in either (basically, the main physics result is given by the active speed fluctuation, which I do not find particularly convincing, see below and the main biology result is that there is no bias towards host cells, the latter analysis could have been done without much of the rest of the paper, so the amount of biology results is quite limited as well), but together this gives an interesting study. Is it important enough for a high impact journal? In my opinion, it is borderline.

If the editor wants to go ahead with this paper, I note a few comments that in my opinion should be considered during a revision.

- 1) Figure 1 and corresponding text: I assume the analysis is done for a single trajectory/single bacterium to avoid effects due to the variations in Ω . If so, this should be stated clearly somewhere. Are results for different bacteria similar?
- 2) What is the accuracy of the speed measurements? Are the observed speed fluctuations clearly larger than the measurement noise? This is crucial for the model.
- 3) p. 6, line 7: The authors note "Interestingly... even in the absence of moving direction fluctuations the MSD grows with time". This is not really surprising.
- 4) The authors emphasize the importance of active speed fluctuations, looking at fig. 3a,b one can however see that most data is for Ω values between -1 and 0, where the speed contribution of speed fluctuations is very small. I am wondering if a model without speed fluctuations would not give an almost equally good fit here. For the smallest Ω , there would be a deviation, but the diffusion coefficient scatters by a factor 2 or 3 there as well.
- 5) p.9 last line, SB-NC and SB-C should be SB-NB and SB-B.
- 6) Notation: b_i in the main text vs $b_{_i}$ in the Supp. Information
- 7) Absence of biased motion: I think a statistical test that positive and negative speeds are equally likely would be useful here.
- 8) Fig. 5: The mean first passage time as a function of Ω would be helpful (as a Supporting figure?).
- 9) Again, the authors stress the importance of speed fluctuations, but in fig. 5f, they chose parameters to maximize their impact and result in a 15% effect.
- 10) Fig.5: Are error bars in c,d, f,g errors of the mean or standard deviation? I think standard deviation would be more informative.

Reviewer #3 (Remarks to the Author):

The manuscript reports the results of a combined theoretical and experimental study of the motility of pathogenic bacteria (*Salmonella*) in the presence of host cells, the statistics of encounter times between bacteria and host cells, and the infection rate. Experiments are performed with various *Salmonella* strains -- wild-type, non-chemotactic, and non-flagellated. The theoretical analysis is based on chiral active particles with active speed fluctuations.

The combined study reveals several interesting results: (i) bacterial speed fluctuations significantly exceed thermal expectations, (ii) a large range of inter-individual variability of the bacterial diffusion coefficients, mainly determined by the range of rotation frequencies and fluctuations in the direction of propulsion, (iii) bacteria do not exhibit biased motion toward host cells, and (iv) bacteria display search time statistics consistent with a random search strategy.

These results are very interesting, and may have important implications for bacterial infection process.

The following comments and questions should be addressed:

- (1) p.5: The authors state that speed fluctuations are not necessarily of thermal origin, but resulting from the motility machinery. At this point in the manuscript, this statement doesn't mean much. It is quantified and discussed later on p.8, and there it seems to be more appropriate.
- (2) p.5/6: The existence of spirals of the average trajectory has been discussed in Ref. [29].
- (3) p.6, bottom: How is the equation for $D_{AV}(\Omega)$ derived from Eq.(5)?
- (4) Figure 2(c-d) seems to indicate that the main dependence of the diffusion coefficient D arises from variations in Ω and D_{θ} , whereas the dependence of D_v is much weaker. Please comment.
- (5) Figure 3(a,b) shows a very strong dependence of D on Ω . This raises the question how the bacteria can vary their rotation frequencies by so much, even changing the sign!
- (6) In Fig. 3(c), couldn't the guide to the eye just as well be a straight line?
- (7) p.10, bottom: The dependence $D(\Omega) \sim 1/\Omega^2$ in Fig. 2(d) and Eq.(5) indicates $\langle \tau \rangle \sim \Omega^2$ ("qualitatively"). Is such a dependence consistent with the observations?
- (8) p.12: What kind of anomalous behavior is found for higher concentrations?
- (9) p.12: The authors state that the NIB (number of infecting bacteria) increases as $1/\langle \tau \rangle$. Isn't this a rather trivial statement, as there is a finite probability for infection at each encounter?
- (10) Is it important that there is a large inter-individual virulence variability? Wouldn't this imply that there is natural selection of bacteria with large D values?
- (11) In Fig. 5(f), there is a small difference of NIB with and without speed fluctuations. Is this difference significant enough to show that speed fluctuations play an important role in virulence?

Reply to Reviewer #1

Otte and colleagues have combined real time imaging, rigorous analysis of the bacterial tracks and modeling to study Salmonella swimming movement on solid surfaces and how this affects the infection process. In particular the modeling approach yields exciting new information about bacterial near surface swimming behavior. The spiral-like movement pattern and its dependence on speed fluctuations, directional fluctuations and the rotation speed provide important new angles for the field. Salmonella mutants are used to assess the role of chemotaxis and motility. These data are nicely in line with previous work and verify that in tissue culture infections, chemotaxis is not contributing to the host cell finding/infection. The present paper significantly extends the original observations by showing mathematically and by simulations that this pertains to both, directionality and search time. Overall, this is a very interesting and timely piece that will be of significant interest to the community. My comments may help the authors to improve the impact, i.e. in the infection biology community.

We would like to thank the reviewer for her/his detailed report and comments that have substantially helped us to improve the manuscript, specially to improve the biological contextualization of the obtained results. We have implemented all suggestions, including all references and corresponding discussions, provided by the reviewer.

Below, we address point-by-points all the comments raised by the reviewer.

1. Fig. 1. It is surprising, that the authors do not show any non-flagellated (i.e. non-motile) cells, which are normally present in vitro and in vivo (PMID: 19096504 and PMID: 1824121). Indeed, in later Figures, such bacteria appear to exist also in their experiments. If omitted on purpose in Fig. 1, this should be explained to the reader.

Fig.1 (panels a, b, c) contains both, motile and non-motile bacteria as in Fig. 4. The small size of the images in Fig.1 makes difficult to identify all bacteria present in the panels (though they are clearly visible by zooming up in the image). The quality of the figure has been improved. Furthermore, in the supplementary video S1, motile and non-motile bacteria can be easily observed. In the panels a, b, and c of Fig. 1, we picked a representative trajectory of a motile bacterium, but certainly there are more bacteria present, including motile and non-motile. We have clarified that the system possesses both, motile and non-motile bacteria, see new caption of figure 1 and text, and refer the two references provided by the reviewer.

2. Page 5. I wonder, if the fluctuations in speed may root in the fluctuations in motor assembly (PMID: 29971711 and PMID: 19234112). Also, previous data had shown fluctuations in flagellar motor speeds (PMID: 2407857 and PMID: 19234112). The authors may want to discuss this with relation to their data.

This is a very interesting question. We showed, by comparing experimental measurements and a series of mathematical motility models that assume thermal fluctuations, that the measured speed fluctuations are not consistent with the hypothesis that fluctuations are of thermal origin. Moreover, we showed that experimentally measured fluctuations are systematically larger than the ones

expected for thermal fluctuations. This led us to speculate that speed fluctuations are of biological origin, and thus likely to be related to fluctuations of the motility machinery. The interesting references mentioned by the reviewer – which we cite in the revised version of the manuscript – deal basically with fluctuations of the flagellar motor at the level of one flagellum. Though certainly of interest to understand speed fluctuations, one has to consider that in bacterial swimming several flagella are involved. It is still unclear how fluctuations at the level of one flagellum affects the temporal dynamics of the bundle and thus how it translates into speed fluctuations. This remains an open question, but certainly it is beyond the scope of our manuscript.

In the revised version of the manuscript, we discuss the relation of fluctuation in the flagellar motor with those of bacterial swimming speed in section 2. “origin of fluctuations” as well as in the discussion section.

3. The abstract, the last paragraph of the introduction and the discussion section indicate that chemotaxis is not essential for the pathogen's interactions with glass surfaces or tissue culture cells. It should be pointed out to the readers, that this has been discovered, before (reference 1). Moreover, this also represents a key limitation of the *in vitro* plus modelling approach. In the gut, chemotaxis is indeed required to promote the infection (PMID: 18241212 and PMID: 15213159). For the infection biology field, it would be helpful to discuss the relevance of the observations for the *in vivo* infection.

In the revised version of the manuscript, we have clarified that it has been reported before that chemotaxis is not essential for the pathogen's interactions on biotic and abiotic surfaces, while it plays a key role in the infection process *in vitro*. We agree with the reviewer that this represents a limitation of the approach. However, we believe that *in vitro* experiments allow quantitatively addressing crucial questions that are of importance in the context of the initial phases of the infection *in vivo*.

In the abstract, it is indicated that chemotaxis related results were reported before, and in the introduction, we included the following comment:

We conclude that ST does not exhibit chemotaxis towards HC, in line with previous studies [PMID: 22911370] that found that chemotaxis is not essential for the pathogen's interaction on glass surfaces or tissue culture cells, but it is required to promote the infection in the gut [PMID: 15213159, PMID: 18241212].

In the discussion section, we added the comment:

[...] the performed chemotaxis statistical tests indicate that ST does not exhibit biased motion towards cells, in line with previous studies [PMID: 22911370] that showed that *in vitro* chemotaxis is not essential for interaction with HCs. Nevertheless, note that *in vivo* chemotaxis is required to promote the infection in the gut [PMID: 15213159, PMID: 18241212].

Finally, we discuss the implications of our study in the context of the in vivo infection also in the discussion section. This point is in relation to point 4 below.

4. A recent study has shown that in vivo, the pathogen in fact glides on the surface of cecum epithelial cells and on the surface of the gel-like mucus layer that covers large parts of the gut epithelium. On the mucus, the proposed random search strategy will actually help the pathogen to "find" breaches in the mucus layer, to reach the underlying epithelium (PMID: 31141690). I wonder, if this may be the main in vivo function of the mechanism described in the present work.

We agree that in order to cross the mucus layer, bacteria most likely make use of a random search strategy, at least during the early stages of infection. In that sense, the unveiled random search strategy provides an insight on how bacteria find these breaches.

On the other hand, it is important to keep in mind that physical properties of mucus may introduce effects/interactions that we have not considered in our study.

In the revised version of the manuscript, we discussed, as suggested by the reviewer, that finding breaches in the mucus layer is a potential function for the described mechanism. However, we would like to point out that in order to corroborate this hypothesis, further experiments would be required.

In the revised version of the manuscript, we refer to PMID: 31141690 also in the introduction, in relation to the role of active motility in the infection process, as well as in the discussion section, as commented above, in the context of random search strategy.

5. The original publication describing the *Salmonella* strains used should be cited (i.e. DOI: 10.1128/IAI.72.7.4138-4150.2004).

The original publication is now cited in the Method section as well as in the introduction.

6. Fig. 3. In the gut, there is not only diffusing bacteria, but also some, that are indeed stuck to the mucus or the surface of epithelial cells (PMID: 31141690). How do the in vitro parameters fit to the *Salmonella* swimming parameters observed on intact gut tissue? Are any of the observed bacteria plotted in Fig. 3 stuck in a similar fashion? The authors may want to discuss how such an adhesion-mediated effect may affect the interpretation of the movement patterns with respect to flagellar function, alone.

It is important to stress that Fig. 3 displays data corresponding exclusively to flagellar-driven motile bacteria. Interestingly, the obtained values for the diffusion coefficient of actively moving bacteria go from very small values, close to those expected for diffusion by Brownian motion (as expected for non-flagellated bacteria), to those for swimming bacteria as reported in PMID: 31141690. This suggests that the developed mathematical model is quantitatively consistent with

data reported on the colon mucus in PMID: 31141690, Fig 1H.

We have added a discussion of our results on light of PMID: 3114169 in subsection inter-individual variability.

7. The beginning of the discussion presents non-directional motility as an infection search strategy as an original discovery from this work. This should be put into context with previous work that has already demonstrated this experimentally (PMID: 22911370).

We have fully rewritten the discussion section, making clear that in PMID: 22911370 it was reported that *in vitro* experiments chemotaxis is not required for the infection process, and we also discussed PMID: 31141690 in relation to the infection process *in vivo*. These works, PMID: 22911370 and PMID: 31141690, are also discussed in the introduction.

Reply to Reviewer #2

This paper presents an experimental and theoretical study of the swimming of *Salmonella typhimurium* near glass surfaces and near surfaces of host cells. The authors first provide a quantitative characterization of the swimming and then ask whether host cells are reached (and subsequently invaded) randomly or by some type of directed/biased motion. Their analysis shows that motion to reach host cells is random, but dependent on motility.

Overall, this is a nice study, which tries to balance interesting physics with interesting biology. I don't think it is a breakthrough in either (basically, the main physics result is given by the active speed fluctuation, which I do not find particularly convincing, see below and the main biology result is that there is no bias towards host cells, the latter analysis could have been done without much of the rest of the paper, so the amount of biology results is quite limited as well), but together this gives an interesting study. Is it important enough for a high impact journal? In my opinion, it is borderline.

If the editor wants to go ahead with this paper, I note a few comments that in my opinion should be considered during a revision.

First of all, we would like to thank the reviewer for her/his detailed report and comments, several of which have helped us to identify points that needed to be clarified or improved.

We believe that our work goes beyond the two points picked up by the reviewer as our only contributions worth mentioning. Our main goal was to arrive to a quantitative description -- implying setting up a reliable mathematical model -- of bacterial motion in the absence and presence of host cells in order to develop a solid mechanistic understanding of the implications of motility in the initial phases of the infection process. With this purpose in mind, we investigated the following aspects of the process. (i) We study the inter-individual variability of motility parameters in the same bacterial population of flagellar driven bacteria. (ii) We perform, to the best of our knowledge, the first assessment of the role of active speed fluctuations in context of the infection process. (iii) We characterize bacterial motion in the presence of host cell and found that (a) there is no chemotaxis, and moreover (b) motility parameters are not affected by their presence. (iv) We performed to the best of our knowledge the first quantification of the search time statistics of these chiral active swimmers. (v) With this knowledge in hand, we developed the first model of the initial phase of the infection process that relates bacterial motility parameters and number of invading bacteria (NIB).

We would like to clarify that one of the aspects of our study was to assess – for the first time -- the role of active speed fluctuations in the context of bacterial infection, but not to over-emphasize their importance. We got the impression that the reviewer interpreted our study assuming that our intention was to over-emphasize speed fluctuation, while we just assessed their impact and implication in the

infection process.

Below, we address point-by-point all technical questions raised by the reviewer.

1) Figure 1 and corresponding text: I assume the analysis is done for a single trajectory/single bacterium to avoid effects due to the variations in Ω . If so, this should be stated clearly somewhere. Are results for different bacteria similar?

The panels (a-c) display three different trajectories, each corresponding to a different value of Ω . The correlations and speed statistics shown in Fig. 1, panels e), f), g) and h), correspond to the trajectory shown in panel (a).

In the new caption of figure 1 this has been clarified.

Note that new figure 1 contains a new panel (panel e) in the current version.

Comparison of trajectory parameters, including their dependency with Ω , are provided in the SI, see Fig. S3.

The variability of motility parameters (v , $D_{\{\theta\}}$, etc), though certainly present (again we refer the reviewer to the SI), is not too big. However, this relatively small variability in parameter values is enough to lead to large variability of the diffusion coefficient D given the nonlinear functional form of D with motility parameters.

2) What is the accuracy of the speed measurements? Are the observed speed fluctuations clearly larger than the measurement noise? This is crucial for the model.

We thank the reviewer for pointing out for this issue. This is undoubtedly a very important point that we have clarified in the new version of the manuscript.

We have included a detailed discussion on this in the SI and clarified this point in the main text by including few comments on this issue.

The measurement noise, as explained in detailed in the SI, is substantially smaller than the observed fluctuations. To illustrate it, we have included a new panel in Fig. 1 (panel e) that displays the speed of the bacterium as function of time, while showing the error bars corresponding to the measurement uncertainty.

Furthermore, we have used estimates of the average speed and associated fluctuations – for details, see equations S18, S19, and S20 – by varying the time interval: $n \Delta t$, with n in the range 1, 2, 3, ... 7, with Δt the inverse of the acquisition frequency (of the order to 1/30 seconds). While certainly we cannot go down to smaller time intervals that the one given by the inverse of the acquisition

frequency, we can use larger time intervals to estimate the average speed. Note that this technique allows us to ensure that we are below the experimental error. We stress that the resulting speed fluctuations are always larger than then one expected by thermal fluctuations.

Said that, we highlight that the experimental error is smaller than fluctuations even when using $n=1$. Obtaining the same resulting using $n>1$ indicate that our results are robust.

3) p. 6, line 7: The authors note “Interestingly... even in the absence of moving direction fluctuations the MSD grows with time”. This is not really surprising.

After the referee’s comment, we have removed “interestingly” from the sentence.

4) The authors emphasize the importance of active speed fluctuations, looking at fig. 3a,b one can however see that most data is for Omega values between -1 and 0, where the speed contribution of speed fluctuations is very small. I am wondering if a model without speed fluctuations would not give an almost equally good fit here. For the smallest Omega, there would be a deviation, but the diffusion coefficient scatters by a factor 2 or 3 there as well.

We would like to clarify that in our study we investigate, among other things, the role of active speed fluctuations during near swimming (when bacterial motion is mainly chiral). One of our goals was to investigate its role at the level of transport coefficients as well as in the infection process, but not to emphasize the importance of active speed fluctuations. To assess the relevance of active speed fluctuations, in Fig. 3, panel b [panel also present in the original submission of the manuscript], we explicitly compare the contribution of active speed fluctuations to the diffusion coefficient. Note that for active non-chiral particles, active speed fluctuations (in the order of the ones reported in experiments) lead to a negligible contribution in the diffusion coefficient. This observation provides a solid argument to justify that in previous studies of bacterial swimming (and considering that most of them focused on bacterial swimming far away from surfaces, i.e. non-chiral swimming), speed fluctuations have been systematically ignored. However, here we show that near surfaces, when bacterial swimming patterns are chiral, active speed fluctuations can lead – though they do not necessarily do -- to a significant contribution to the diffusion coefficient. This depends, as noticed by the reviewer, on the value of Omega, and for Omega values (in absolute value) larger than 1, using the approximation of the “average bacterium” [red curve in panels a and b of Fig. 3], the contribution of active speed fluctuations to the diffusion coefficient is not negligible, while below values of 1, in this approximation, is negligible.

The (quantitative) answer to the reviewer’s question is given by panel b of Fig. 3 that provides an explicit measurement of the contribution of active speed fluctuations. In 9% of the trajectories, active speed fluctuations contribute to more than 10% of the actual value (up to 40%), and in 34%, the contribution is larger

than 1% of the value.

5) p.9 last line, SB-NC and SB-C should be SB-NB and SB-B.

We have modified it. We thank the reviewer to point it out.

6) Notation: b_i in the main text vs b_i in the Supp. Information

We have modified it. We thank the reviewer to point it out.

7) Absence of biased motion: I think a statistical test that positive and negative speeds are equally likely would be useful here.

In the new version of the manuscript, we included a binomial test to evaluate whether positive and negative speeds are equally likely as suggested by the reviewer. The statistical test indicates that experimental data is consistent with absence of a chemotactic response.

8) Fig. 5: The mean first passage time as a function of Ω would be helpful (as a Supporting figure?).

We agree that this would be a very informative figure. We have now included the figure in SI (see Fig. S7).

9) Again, the authors stress the importance of speed fluctuations, but in fig. 5f, they chose parameters to maximize their impact and result in a 15% effect.

Our intention was not to emphasize the importance of speed fluctuation, but to assess their impact on the infection process. This is the first study, to the best of our knowledge, that includes speed fluctuations in this context. Panel 5f, corresponding to $\Omega=2.5$, aims at illustrating and understanding how the underestimation of the diffusion coefficient (by neglecting active fluctuations) impacts the infection process. For $\Omega>1$ there exists a statistically significant increase due to speed fluctuations. Within the experimentally measured values of Ω , we find that for $\Omega>2.5$ speed fluctuations can account for 15% of the infection capacity. We do not believe that a different of 15% is negligible corrections [for instance, the consequences of a 15% correction in the basic reproduction number R_0 can be very important, and while here we are not computing a R_0 , from NIB we can estimate the number of second infections].

In the SI we have included a new figure, Fig. S10, that shows results for $\Omega=0, 1, 2, 3, 4, \text{ and } 5$. We also performed a statistical test that indicates that speed fluctuations lead to a statistically significant increase in the number of infected bacteria for all $\Omega\geq 1$ (see Supplementary Note 8). In the main text, we clarify that we are performing just an assessment of the role of speed fluctuations and

illustrating results with $\Omega=2.5$, while a larger range of Ω values is shown in the SI.

10) Fig.5: Are error bars in c,d, f,g errors of the mean or standard deviation? I think standard deviation would be more informative.

The error bars in (c), (f), and (g) correspond to the standard error of the mean (SEM) for $\langle \tau \rangle$ and standard error (SE) for NIB. Note that since we are dealing with exponential distributions for τ , for which the standard deviation is equal to the mean, the standard deviation will not be informative.

Reply to Reviewer #3

The manuscript reports the results of a combined theoretical and experimental study of the motility of pathogenic bacteria (*Salmonella*) in the presence of host cells, the statistics of encounter times between bacteria and host cells, and the infection rate. Experiments are performed with various *Salmonella* strains -- wild-type, non-chemotactic, and non-flagellated. The theoretical analysis is based on chiral active particles with active speed fluctuations.

The combined study reveals several interesting results: (i) bacterial speed fluctuations significantly exceed thermal expectations, (ii) a large range of inter-individual variability of the bacterial diffusion coefficients, mainly determined by the range of rotation frequencies and fluctuations in the direction of propulsion, (iii) bacteria do not exhibit biased motion toward host cells, and (iv) bacteria display search time statistics consistent with a random search strategy.

These results are very interesting, and may have important implications for bacterial infection process.

We would like to thank the reviewer for her/his positive, detailed report and comments that have substantially helped us to improve the manuscript. We have implemented all suggestions made by the reviewer.

Below, we address point-by-points all the comments raised by the reviewer.

The following comments and questions should be addressed:

(1) p.5: The authors state that speed fluctuations are not necessarily of thermal origin, but resulting from the motility machinery. At this point in the manuscript, this statement doesn't mean much. It is quantified and discussed later on p.8, and there it seems to be more appropriate.

We followed the referee suggestion and rewrote the sentence in question.

(2) p.5/6: The existence of spirals of the average trajectory has been discussed in Ref. [29].

We agree that in Ref. [29] it has been shown that $\langle x \rangle(t)$ displays an inward spiral trajectory. However, here we find that due to active speed fluctuations, and in the absence of angular fluctuations, $\langle x \rangle(t)$ takes the form of an outward spiral that converges to a limit cycle. This was not reported in Ref. [29].

In the revised version of the manuscript, we clarified that the inward spiral trajectory of $\langle x \rangle(t)$ has been reported previously in Ref. [29].

(3) p.6, bottom: How is the equation for $D_{AV}(\Omega)$ derived from Eq.(5)?

The $D_{AV}\{\Omega\}$ correspond to the diffusion coefficient of a representative bacterium whose motility parameters are the average motility parameter obtained

in experiments, except for Ω . This is because we understood that the nonlinear form of the diffusion coefficient with Ω makes it particularly sensitive to variations in Ω , while it is less sensitive to variation in the other parameters. However, there is a caveat: we use $\langle D_v/k_v^2 \rangle$ since estimates of D_v/k_v^2 are more reliable than the estimations of D_v and k_v taken independently.

(4) Figure 2(c-d) seems to indicate that the main dependence of the diffusion coefficient D arises from variations in Ω and D_θ , whereas the dependence of D_v is much weaker. Please comment.

The diffusion coefficient D strongly depends on Ω , but other motility parameters are also important. In particular, as Ω is increased, the contribution of speed fluctuations to the diffusion coefficient becomes increasingly important. Moreover, for Ω values around 3, 5, 10 or larger, the estimates of D neglecting speed fluctuations are catastrophic, since speed fluctuations dominate the diffusion process. We have added a comment at the end of the section to clarify this issue.

(5) Figure 3(a,b) shows a very strong dependence of D on Ω . This raises the question how the bacteria can vary their rotation frequencies by so much, even changing the sign!

This is indeed a very interesting question. While unfortunately we ignore how in our experiments bacteria regulate their rotation frequency, we know that i) by controlling the flagellar-bundle rotation frequency, ii) by synthesizing and displaying on the bacterial surface different adhesins (that strongly affect the rotational and translation drag coefficients), or iii) by regulating its aspect ratio, bacteria can affect the rotation frequency Ω .

(6) In Fig. 3(c), couldn't the guide to the eye just as well be a straight line?

We could have used a straight line, but the fitting is significantly worse. It seems that Ω saturates towards small positive values, as in aspect ratio is increase. That is why we chose to use a non-linear fitting. Anyway, we insist that this is a guide to the eye.

(7) p.10, bottom: The dependence $D(\Omega) \sim 1/\Omega^2$ in Fig. 2(d) and Eq.(5) indicates $\langle \tau \rangle \sim \Omega^2$ ("qualitatively"). Is such a dependence consistent with the observations?

One expects the regime $D(\Omega) \sim 1/\Omega^2$ to emerge for large values of Ω . However, as Ω increases, speed fluctuations become increasingly important. For large values of k_v (much larger than D_θ and Ω), the contribution of speed fluctuations to the diffusion coefficient is a constant. In experiments, we have verified the relation $\langle \tau \rangle = 1/D(\Omega)$, see new Fig.5(d).

In the SI, we included a new figure that shows D as function of Omega (see Fig. S7).

(8) p.12: What kind of anomalous behavior is found for higher concentrations?

In higher concentrations, it has been reported super-diffusive behavior. The sentence has been rewritten to clarify this point.

(9) p.12: The authors state that the NIB (number of infecting bacteria) increases as $1/\langle\tau\rangle$. Isn't this a rather trivial statement, as there is a finite probability for infection at each encounter?

We agree that it is a direct consequence of the proposed model. However, it is less trivial than what it seems. As the probability of infection decreases, the quantity that controls NIB is rather the frequency of repeated encounters, which is no longer the mean-first passage time. Beyond this rather mathematical observation, in the context of bacterial infection, our model represents the first mechanistic explanation that relates motility parameters and infection capacity. Let us recall that so far there is no quantitative mechanistic explanation to correlate motility parameters and infection capacity (NIB). Note that predictions resulting from this mechanism could be tested experimentally.

(10) Is it important that there is a large inter-individual virulence variability? Wouldn't this imply that there is natural selection of bacteria with large D values?

A large interindividual virulence variability may represent an adaptive bacterial response to cope with difference biological context. In that respect, we believe it is biologically a very important finding. In this particular *in vitro* experiment, it is tempting to believe that natural selection would favor large D values. However, it is likely that bacteria need a large pool of D values to exhibit an efficient response in a large variety of contexts, including some where small D values would be advantageous. The questions raised by the referee are very interesting, but also very complex. It is out of the scope of this study.

(11) In Fig. 5(f), there is a small difference of NIB with and without speed fluctuations. Is this difference significant enough to show that speed fluctuations play an important role in virulence?

In the new version of the SI we included a figure that compares the values of NIB in simulations with speed fluctuations and without speed fluctuations for $\Omega=0, 1, 2, 3, 4, \text{ and } 5$ (see Fig. S10). We have also performed statistical tests to assess the statistically significant of the results. We found that the increase of NIB due to speed fluctuations is statistically significant for values of $\Omega \geq 1$ (see Supplementary Note 8). In short, the difference observed in Fig. 5(f) is statistically significant.

Reviewer #2 (Remarks to the Author):

In my opinion, the authors have done an excellent job in their revision and clarified all issues raised by the reviewers. I particularly like the new fig 1e that compares the speed fluctuations with the measurement error. Likewise, the discussion of the biological relevance in the infection process has been improved and the statistical tests give more weight to the observed lack of chemotaxis. I recommend the paper's acceptance.

Reviewer #3 (Remarks to the Author):

In their resubmittal letter, the authors have responded in detail to all points raised in my previous report. They have modified their manuscript accordingly. This is a manuscript which reports many interesting results, and I support its publication in the present form.